# Estimating microbial population data from optical density

**Portia Mira** [1] *, **Pamela Yeh** [1,2], **Barry G. Hall** [3]

**1** Department of Ecology and Evolutionary Biology, University of California, Los Angeles, Los Angeles, CA, United States of America, **2** Santa Fe Institute, Santa Fe, New Mexico, United States of America, **3** Bellingham Research Institute, Portland, OR, United States of America

* portiamira20@gmail.com

## Abstract

The spectrophotometer has been used for decades to measure the density of bacterial populations as the turbidity expressed as optical density–OD. However, the OD alone is an unreliable metric and is only proportionately accurate to cell titers to about an OD of 0.1. The relationship between OD and cell titer depends on the configuration of the spectrophotometer, the length of the light path through the culture, the size of the bacterial cells, and the cell culture density. We demonstrate the importance of plate reader calibration to identify the exact relationship between OD and cells/mL. We use four bacterial genera and two sizes of micro-titer plates (96-well and 384-well) to show that the cell/ml per unit OD depends heavily on the bacterial cell size and plate size. We applied our calibration curve to real growth curve data and conclude the cells/mL–rather than OD–is a metric that can be used to directly compare results across experiments, labs, instruments, and species.

**Data Availability Statement:** All relevant data are within the paper and its Supporting Information files.

**Funding:** We are grateful for funding from a KL2 Fellowship (PJY) through the NIH/National Center for Advancing Translational Science (NCATS) UCLA

## Introduction

The Beer-Lambert law [1] relates the molar concentration (C) of a solute to absorbance of light according to the equation $C = \epsilon A$ where $\epsilon$ is the molar extinction coefficient and A is the absorbance. Epsilon ($\epsilon$) is given at a specific wavelength and specific light path, usually a 1 cm light path. That relationship is what allows us to monitor enzyme reactions by absorbance, to measure protein concentrations by absorbance, and to do enzyme-linked immunoassays (ELISA).

However, the Beer-Lambert law applies only to solutions in which molecules of solute are uniformly distributed throughout the solvent. It does not apply to suspensions of particulate matter such as microbial cells. Rather than absorbing light, particles scatter light, which is why we express turbidity as OD (optical density) instead of A (absorbance). The relationship between cells/mL and OD is complex and depends on several factors including length of light path, size of the particles (cells), and number of particles. There is no simple factor equivalent to $\epsilon$ that relate number of cells/mL to OD.

It is not a trivial matter to determine the number of cells/mL or the mass of cells in a culture. The classic way was to dry a culture and weigh the cells, a method that does not lend itself to easy measurement of cell densities in small cultures, (to say nothing of the fact that while weighing the dehydrated cells, they absorb moisture from the air and the weight increases even

CTSI Grant Number UL1TR001881, the Presidential Postdoctoral Fellowship and the Ruth L. Kirschstein National Research Service Award (AI007323) to Dr. Portia Mira The funders had no role in the study design, data collection and analysis, decision to publish or prepare the manuscript.

**Competing interests:** The authors have declared that no competing interests exist.

as the balance is watched). It can be important to determine cell densities easily and quickly, i.e., when monitoring growth in fermenters to determine when to harvest cells.

The convenience of measuring cell populations in microtiter plate readers led us to determine the relationship between OD and cells/mL for several microbial species and for plates of different sizes. Given that relationship OD can be used to calculate cell numbers just as A is used to calculate concentration of a solute.

Spectrophotometers have been used for over 6 decades as a means of measuring the population density of microbial cultures [2–4]. Population density is estimated from the turbidity of the culture and is typically expressed as OD (optical density), typically at a wavelength of 600 nm. OD is the negative log of transmittance, which is the fraction of the light that is detected when it is passed through a cuvette that contains a sample of the culture. The Beer-Lambert law states that OD is proportional to the concentration of a solution [1]. However, this law does not apply to suspensions of particles (or bacterial cultures) because instead of absorbing light, light is scattered off the axis of the detector [5, 6]. As a result, the OD is proportional to the cell titer only up to a limited point, typically an OD of about 0.1 (Fig 3). Above that range, some of the light that is scattered away from the detector by one cell is subsequently scattered back to the detector by another cell [7]. As a result, the OD does not increase as fast as does the cell titer and therefore, one cannot rely on OD alone to accurately measure bacterial population densities.

To precisely estimate cell titers from observed OD measurements, it is necessary to calibrate the spectrophotometer. The relationship of OD to cell titer depends on four components: 1) the configuration of the spectrophotometer, 2) the length of the light path through the suspension, 3) the size of the cells, and 4) the cell culture density. Therefore, it is necessary to calibrate each spectrophotometer model separately for each microbial species that is to be studied.

Until about a decade ago ODs were determined by putting a sample of the culture into a cuvette of, typically, a 1 cm light path. Determining the growth rate required sampling from a culture at timed intervals and recording the OD at each time point. In practical terms it was difficult to follow more than about 20 cultures simultaneously. The advent of using a microtiter plate reader to monitor the growth of cultures in the wells of a microtiter plate permits high throughput measurements of microbial growth kinetics. However, the same considerations of calibration apply to microtiter plate readers as to spectrophotometers [7]. Microtiter plates have various sizes (i.e., 96-wells, 384-wells) which means each well has different depths. Therefore, it is necessary to calibrate a plate reader separately for each size plate.

A recent study shows the benefit of plate reader calibrations using silica microspheres [8]. However, they focus their study on only *E. coli* and do not consider microtiter plate size, well-depth, or other sizes of bacterial species. Here, we demonstrate the importance of calibrating a plate reader using a Biotek Epoch 2 plate reader, both 96-well and 384-well microtiter plates and four bacterial species that span a wide range of cell sizes. We then apply the calibration to a set of growth curves for *Escherichia coli* and show that using cells/mL yields the same growth rates as using OD.

## Materials and methods

### Bacterial strains

We used four bacterial strains of different genera: *Escherichia coli* K12 strain DH5α (F– φ80lacZΔ M15 Δ (*lacZYA-argF*) *U169 recA1 endA1 hsdR17* (rK–mK+) *phoA supE44 λ- thi–1 gyrA96 relA1*) from ThermoFisher, *Escherichia coli* strain CFT073 O6:K2:H1 [10], *Pseudomonas putida* strain ATCC 12633, *Staphylococcus epidermidis* strain ATC12228 [11] and *Bacillus megaterium* strain ATCC 14581.

## Plate reader calibration

To identify the colony forming units per genus, we inoculated four standing overnight cultures for each genus. Cultures were inoculated at 37°C in 10mL of LB (10 g tryptone, 5 g yeast extract, 10 g NaCl per liter) and placed in 15mL culture tubes with tightly sealed caps that allowed no aeration for 16–18 hours. Cultures of each genus were combined into a 50-mL conical tube and spun down at 4,000 rpm for 15 minutes at 4°C then resuspended in 4mL M9 buffer, this led to a 4X concentrated starting bacterial culture. Two sets of fifteen dilutions were made. The first, starting cultures were diluted at concentrations that gave the most countable number of colonies. These were $10^6$, $10^7$, $10^8$ for *E. coli*, and $10^4$ and $10^5$ for *S. epidermidis*, *P. putida*, and *B. megaterium*. Dilutions were plated on LB agar, inoculated at 37°C overnight and counted the following day. The second set of dilutions were 15 two-fold dilutions from the starting culture. Each tube was plated in 96-well plates (4 replicates 200 µl per well) and 384-well (6 replicates, 80µl per well) plus blank wells (only M9) for each plate size. The $OD_{600}$ was measured every 5 minutes for 30 minutes using the Biotech Epoch 2 plate reader. We report the corrected OD, that is the OD of only media subtracted from each experimental reading (raw data can be found in S3 File). Aggregated data was used in combination with the colony counts to obtain the individual calibration for each genus.

The protocol described in this peer-reviewed article is published on protocols.io, dx.doi.org/10.17504/protocols.io.8epv5j6wjl1b/v1 and is included for printing as S1 File with this article. Please see S2 File- "Calibration calculator.xlsx" which facilitates using that protocol and S3 File—"Calibration_RawData.xlsx" for a complete set of raw data.

## Growth rate experiments

Standing overnight cultures of *E. coli* and *S. epidermidis* were diluted (1:20) to obtain a starting OD of 0.02–0.03. Cultures were then plated across the row of a 96-well plate (12 replicate wells) and the $OD_{600}$ was measured every 20 minutes for 22 hours. The growth rates were calculated from the OD measurements using the program GrowthRates [9] version 5.1 (https://bellinghamresearch.com/).

## Results

### Calibration curves

Standing overnight cultures for each organism (*E. coli DH5α*, *S. epidermidis*, *B. megaterium* and *P. putida)* were concentrated to about 2.5 x $10^9$ cells/mL in mineral salts (M9) buffer and 2x serially diluted. Each dilution, plus a buffer blank, was distributed to four wells (96 well plate) or 6 wells (384 well plate) and the ODs were measured (plate layouts can be found in S3 File. For each dilution, the mean OD was corrected by subtracting the mean OD of the blank (buffer) well, and corrected ODs were graphed vs the number of viable cells. Stevenson et al. [7] suggested that a quadratic relationship exists between cell number and OD. However, to identify the best possible fit, we wanted to explore other relationships. We fit curves to the resulting points based on assumption of four relations- a linear relationship, a quadratic relationship, a cubic relationship, and a polynomial of degree 4 relationship. *E. coli* fits are shown as representative data (Fig 1) and the corresponding $R^2$ values, the correlation coefficients of the fits, for the other genera measured are also shown in Table 1.

The linear fit is clearly inappropriate, with $R^2 = 0.95$ for both 96 and 384 well plates. To choose among the other fits, we considered $R^2$ as a measure (Table 1). We found that the $R^2$ criterion for the polynomial of degree 4 fit is the best for *E. coli*.

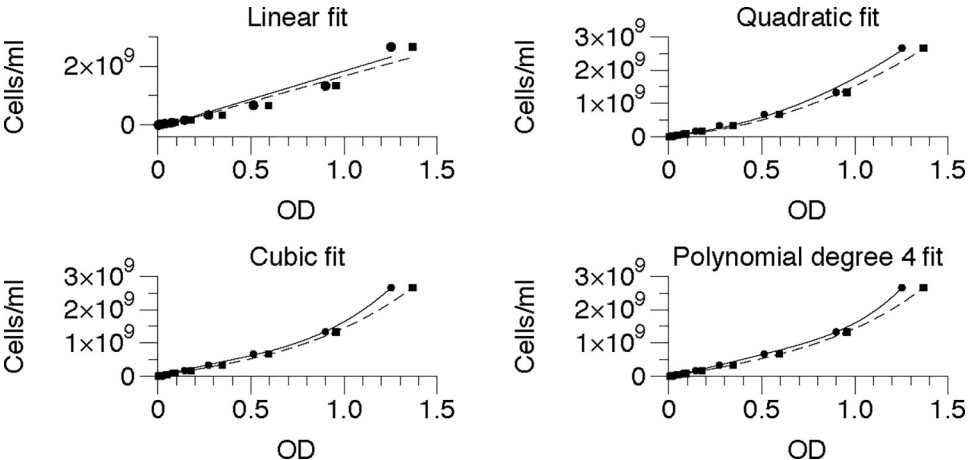

**Fig 1. Relationship between cells/mL and OD with different fits.** The solid lines and round points represent 96-well measurements, and the dashed lines and square points represent 384-well measurements for *E. coli*.

We similarly calibrated the plate reader with *Staphylococcus epidermidis*, *Pseudomonas putida*, and *Bacillus megaterium*. In each case the polynomial of degree 4 was the best fit. The approximate cells/mL with the polynomial of degree 4 using the general equation: A $OD^4$ + B $OD^3$ + C $OD^2$ + D OD + E where A, B, C, D and E are the coefficients of the terms. Table 2 shows the polynomial degree 4 equations for each organism and plate size. We also considered another criterion for goodness of fit, Root Square Mean Error (RMSE) (data shown in S3 File). The smaller is RMSE, the better the fit. By RMSE criterion, polynomial degree 4 was also consistently the best fit.

The equations are different across species and within a species for 96 and 384 well plates (Table 2). This emphasizes the need to calibrate each species and plate size separately. We provide these equations solely as examples, and we emphasize that they should not be used for instruments other than the Biotek Epoch 2.

The cells/mL at an OD of 1 decreases as the cell volume (CV) increases according to a quadratic function in which cells/mL at OD of 1 is equal to 2.1 e8 x $CV^2$ – 5.9 e9 x CV + 4.0 e10, with $R^2$ = 0.998 for 96-well plates and 1.2 e8 x $CV^2$ – 3.4 e9 x CV + 2.3 e10, with $R^2$ = 0.999 for 384-well plates. This is consistent with Koch's 1961 and Stevenson et al.'s 2016 finding [2, 7].

### Application of calibration curve to real growth curve data

The growth of two *E. coli* strains and one *S. epidermidis* strain at 37˚ in LBD medium was monitored. Population density was measured as corrected OD and cells/mL based on a quadratic-fit calibration curve. Fig 2 shows a plot of one well for *S. epidermidis* strain and

**Table 1. $R^2$ values for the different fits for each of the four bacterial genera measured.**

| Organism | 96 well | | | 384 well | | |
|---|---|---|---|---|---|---|
| | Quadratic | Cubic | Polynomial | Quadratic | Cubic | Polynomial |
| *S. epidermidis* | 0.9987 | 0.99995 | 1.0 | 0.99989 | 0.99995 | 0.99999 |
| *E. coli* | 0.99452 | 0.99961 | 1.0 | 0.99815 | 0.99995 | 1.0 |
| *P. putida* | 0.99824 | 0.99996 | 0.99996 | 0.99821 | 0.99987 | 0.99992 |
| *B. megaterium* | 0.99977 | 099999 | 1.0 | 0.99882 | 0.99936 | 0.99997 |

Organisms are listed in order of cell volume.

**Table 2. Calibration equations and cell size.**

| Species | Plate Size | Polynomial Degree 4 Equation, Cells/mL = | Cells/mL @ OD = 1 | Cell Vol |
|---|---|---|---|---|
| *S. epidermidis* | 96 | $4.3e10\ OD^4 - 3.8e10\ OD^3 + 1.2e10\ OD^2 + 1.7e10\ OD + 1.7e8$ | $3.42 \times 10^{10}$ | $1\ \mu^3$ |
| | 384 | $-2.5e10\ OD^4 + 4.2e10\ OD^3 - 1.2e10\ OD^2 + 1.5e10\ OD + 9.4e7$ | $2.01 \times 10^{10}$ | |
| *E. coli* | 96 | $1.6e9\ OD^4 - 2.3e9\ OD^3 + 1.3e9\ OD^2 + 1.0e9\ OD + 5.1e5$ | $1.6 \times 10^9$ | $9.8\ \mu^3$ |
| | 384 | $3.3e8\ OD^4 - 2.1e8\ OD^3 + 4.9e8\ OD^2 + 8.3e8\ OD + 4.2e4$ | $1.44 \times 10^9$ | |
| *P. putida* | 96 | $2.4e8\ OD^4 - 2.7e8\ OD^3 + 6.4e7\ OD^2 + 4.7e8\ OD + 1.4e5$ | $5.04 \times 10^8$ | $12.3\ \mu^3$ |
| | 384 | $3.7e8\ OD^4 - 6.8e8\ OD^3 + 3.8e8\ OD^2 + 3.7e8\ OD + 8.2e5$ | $4.41 \times 10^8$ | |
| *B. megaterium* | 96 | $4.4e8\ OD^4 - 4.9e8\ OD^3 + 3.2e8\ OD^2 + 5.9e8\ OD + 4.3e6$ | $8.64 \times 10^8$ | $17\ \mu^3$ |
| | 384 | $-1.2e9\ OD^4 + 3.6e9\ OD^3 - 2.7e9\ OD^2 + 1.2e9\ OD - 5.3e6$ | $8.95 \times 10^8$ | |

highlights that the curves based on OD and cells/mL are almost identical. For the *S. epidermidis* culture in Fig 2, the growth rate based on OD was $\mu = 0.01459 \pm 0.000412\ \text{min}^{-1}$ based on 6 points from 140 through 240 minutes, with R = 0.9984. Based on cells/mL, the growth rate was similar, $\mu = 0.01354 \pm 0.000285\ \text{min}^{-1}$ based on 6 points from 140 through 240 minutes, with R = 0.9983.

Fig 3 shows a growth curve of *E. coli* based on OD and the same curve based on scaled cells/mL. Above an OD of 0.1 the OD (open circles) is significantly below the scaled cells/mL, illustrating that the proportionality of cells/mL to OD falls off above OD = 0.1.

The program GrowthRates [9] version 5.1 (https://bellinghamresearch.com/) was used to estimate the growth rates in 12 wells for *E. coli* K12 strain DH5, the uropathogenic *E. coli* strain CFT073 [10], and *S. epidermidis* strain. ATCC 12228 [11]. We found the growth rate estimates similar when comparing corrected OD to cell/ml using the polynomial degree 4 fit. The growth rate estimated from cells/mL was significantly different from the growth rate based on OD for *E. coli* CFT073 and *S. epidermidis* (Table 3).

The growth rates estimated from OD and from cells/mL are not the same. Which estimates should we trust more? We trust the rates based on cells/mL because when OD reaches above 0.1, the OD readings fall off as the true population density (cells/mL) increases.

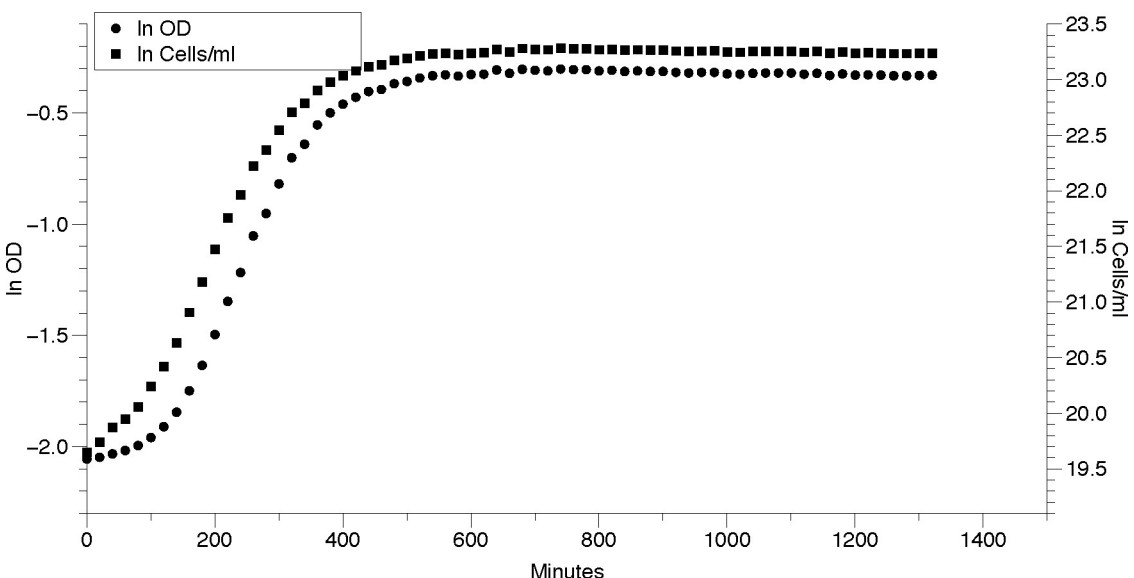

**Fig 2. Growth curves of *S. epidermidis* in one well based on different measures of population density.** The natural log of OD (circles) and cells per mL (squares) is plotted over time (minutes).

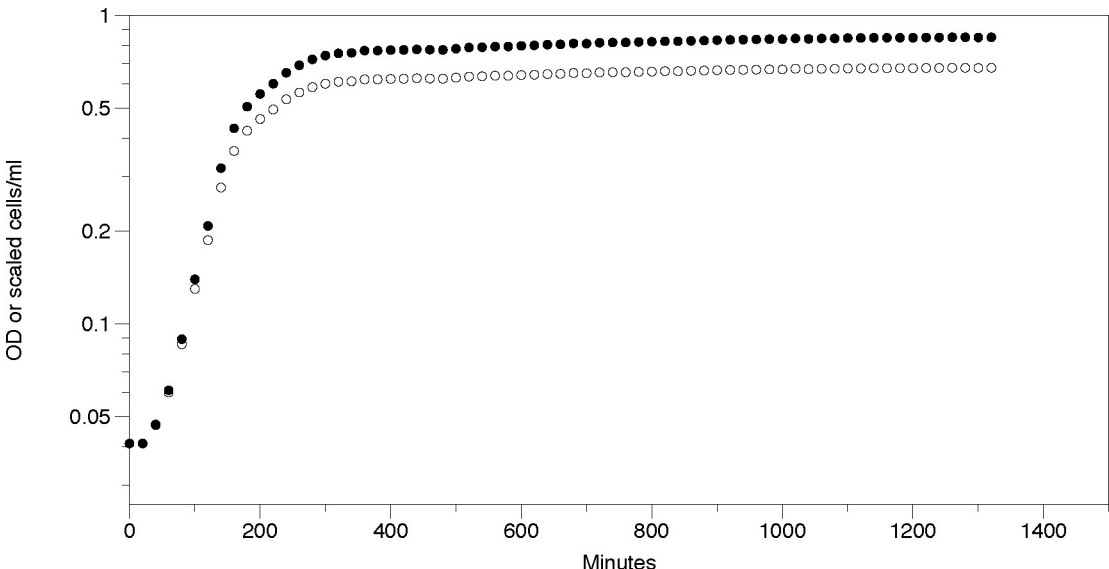

**Fig 3. Growth curves based on OD and based on scaled cells/mL.** The values obtained for cells/mL have been scaled to fit on the same scale as OD by dividing cells/mL by $1.07 \times 10^9$. Curves based on OD are represented by open circles and scaled represented by closed circles.

## Discussion

Our work highlights the importance of calibrating a microplate reader. We use four different bacterial genera to explore the relationships between corrected OD and cells/mL. First, we compared a quadratic, cubic and polynomial degree 4 fit to bacterial growth data and show that for all four genera, the best calibration fit is a polynomial of degree 4 (Table 1). To highlight the importance of calibrating the plate reader separately for 96-well and 384-well plates, we show the differences in the polynomial degree 4 equations. This difference likely arises from the different culture depths, hence different light path lengths, in 96- vs 384- well plates (Table 2). We also emphasize the importance of separate calibrations for each genus (Table 2). The calibration coefficients depend upon the cell volume, with the sum of those coefficient decreasing as a cubic function as the microbial cell volume increases. Good calibration and application of the calibration curve clearly depends upon consistent well volumes, not only within a single experiment, but between experiments.

Growth rates estimated from OD and cells/mL are not identical (Table 3), but we trust the rates estimated from cells/mL more than those estimated from OD.

Why is it worth the effort to calibrate a plate reader? First, because it allows us to express the maximum population density, i.e., the carrying capacity of the medium, in terms of cells/mL rather than OD. Consider the maximum OD for *E. coli* CFT073 and *S. epidermidis* in 96

**Table 3. Comparison of growth rates based on OD and cells/mL.**

|  | *E. coli* K12 DH5α | | | *E. coli* CFT073 | | | *S. epidermidis* ATCC 12228 | | |
|---|---|---|---|---|---|---|---|---|---|
|  | μ | R | max OD or cells/mL | μ | R | max OD or cells/mL | μ | R | max OD or cells/mL |
| OD | 0.0212 | 0.9968 | 0.348 | 0.0235 | 0.9989 | 0.622 | 0.0155 | 0.9989 | 0.610 |
| cells/mL | 0.0207 | 0.9984 | $4.36 \times 10^8$ | 0.0253 | 0.9990 | $8.21 \times 10^8$ | 0.0139 | 0.9986 | $1.25 \times 10^{10}$ |
| p-value | 0.55 | | | 0.0024 | | | 9.19e-6 | | |

Values are means of 12 replicates. In all cases the S.E. was < 0.05 of the mean.

well plates (Table 3). These values are very similar (0.622 and 0.610 respectively), but for *E. coli*, the OD of 0.622 represents only 8.2 x $10^8$ cells/mL. On the other hand, the OD of 0.610 for *S. epidermidis* represents 1.25 x $10^{10}$ cells/mL. This is a fifteen-fold difference in the number of cells per milliliter in each overnight culture. This difference is important to consider when performing experiments that depend on the number of cellular divisions or cells present, such as cellular communication [12, 13] and antibiotic susceptibility [14–17] and biofilms [18].

Knowing the relationship between OD and cells/mL is not just valuable during growth rate determinations. For instance, when monitoring the growth yield in a fermenter it is very valuable to know the actual population density to decide when to harvest the cells. For *S. epidermidis* if the yield according to OD, when the OD = 2.5, that corresponds to 1.2 x $10^{12}$ cells/mL, which is five times the yield when OD = 0.5 (9.5 x $10^9$ cells/mL).

Probably the most important reason to calibrate plate readers is to use a consistent metric for expressing population densities. By expressing population densities in cells/mL, rather than OD, experiments can be directly compared from different instruments, different labs, and even different genera. Our work shows that using cells/mL as a metric permits reliable measurements of growth rates as does using OD (Table 3) because cell/ml allows consistency when expressing population densities. To measure bacterial growth rates more precisely, we encourage all to calibrate their instruments and to express their results in cells/mL. A plate reader calibration protocol (S1 File) and calibration calculator (S2 File) can be found in the supplemental information. All raw data we used to calculate growth rates and calibrate the spectrophotometer can be found in S3 File.

## Supporting information

**S1 File. Plate reader calibration protocol.** PDF file that contains step by step instructions on how to perform a plate reader calibration published on protocols.io.
(PDF)

**S2 File. Calibration calculator.** Excel file that contains pre-labeled cells for a 96-well plate or 384-well plate that will calculate the number of cells per milliliter in a starting culture. User will input the OD values from the plate.
(XLSX)

**S3 File. Calibration raw data.** Excel file that contains the plate layouts and all optical density readings for the organisms used in this protocol.
(XLSX)

## Author Contributions

**Conceptualization:** Barry G. Hall.

**Data curation:** Portia Mira.

**Formal analysis:** Barry G. Hall.

**Funding acquisition:** Pamela Yeh.

**Investigation:** Portia Mira.

**Methodology:** Portia Mira.

**Project administration:** Portia Mira.

**Resources:** Pamela Yeh.

**Software:** Barry G. Hall.

**Supervision:** Barry G. Hall.

**Validation:** Portia Mira.

**Visualization:** Barry G. Hall.

**Writing – original draft:** Portia Mira, Barry G. Hall.

**Writing – review & editing:** Portia Mira, Pamela Yeh, Barry G. Hall.

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
