## [Decision Letter · Decision Letter 0]

8 Aug 2022

PONE-D-22-14390Estimating Microbial Population Data from Optical DensityPLOS ONE

Dear Dr. Mira,

Thank you for submitting your manuscript to PLOS ONE. After careful consideration, we feel that it has merit but does not fully meet PLOS ONE’s publication criteria as it currently stands. Therefore, we invite you to submit a revised version of the manuscript that addresses the points raised during the review process.

We look forward to receiving your revised manuscript.

Kind regards,

Abdelwahab Omri, Pharm B, Ph.D, Laurentian University

Academic Editor

PLOS ONE

Journal Requirements:

“We are grateful for funding from a KL2 Fellowship (PJY) through the NIH/National Center for Advancing Translational Science (NCATS) UCLA CTSI Grant Number UL1TR001881 as well as the Presidential Postdoctoral Fellowship to Dr. Portia Mira”

“We are grateful for funding from a KL2 Fellowship (PJY) through the NIH/National Center for Advancing Translational Science (NCATS) UCLA CTSI Grant Number UL1TR001881 as well as the Presidential Postdoctoral Fellowship to Dr. Portia Mira.”

“We are grateful for funding from a KL2 Fellowship (PJY) through the NIH/National Center for Advancing Translational Science (NCATS) UCLA CTSI Grant Number UL1TR001881 as well as the Presidential Postdoctoral Fellowship to Dr. Portia Mira”

Reviewers' comments:

Reviewer's Responses to Questions

**Comments to the Author**

1. Does the manuscript report a protocol which is of utility to the research community and adds value to the published literature?

Reviewer #1: Yes

2. Has the protocol been described in sufficient detail?

Descriptions of methods and reagents contained in the step-by-step protocol should be reported in sufficient detail for another researcher to reproduce all experiments and analyses. The protocol should describe the appropriate controls, sample sizes and replication needed to ensure that the data are robust and reproducible.

Reviewer #1: Yes

3. Does the protocol describe a validated method?

Reviewer #1: Yes

4. If the manuscript contains new data, have the authors made this data fully available?

Reviewer #1: Yes

**5. Is the article presented in an intelligible fashion and written in standard English?**

Reviewer #1: Yes

6. Review Comments to the Author

Reviewer #1: A good technical manuscript with a few obvious typos. The message nicely confirms well-known concepts in terms of accuracy of OD vs cell count, the influence of cell size, etc.

7. PLOS authors have the option to publish the peer review history of their article (what does this mean?). If published, this will include your full peer review and any attached files.

Reviewer #1: No

---

## [Author Response · Author response to Decision Letter 0]

19 Sep 2022

Journal Requirements:

Done

“We are grateful for funding from a KL2 Fellowship (PJY) through the NIH/National Center for Advancing Translational Science (NCATS) UCLA CTSI Grant Number UL1TR001881 as well as the Presidential Postdoctoral Fellowship to Dr. Portia Mira”

Done – this has been updated in the cover letter above.

“We are grateful for funding from a KL2 Fellowship (PJY) through the NIH/National Center for Advancing Translational Science (NCATS) UCLA CTSI Grant Number UL1TR001881 as well as the Presidential Postdoctoral Fellowship to Dr. Portia Mira.”

“We are grateful for funding from a KL2 Fellowship (PJY) through the NIH/National Center for Advancing Translational Science (NCATS) UCLA CTSI Grant Number UL1TR001881 as well as the Presidential Postdoctoral Fellowship to Dr. Portia Mira”

Done – this has been revised and included at the end of the cover letter above as well as on the online form. 

Raw data has been added as a supplementary file and referenced in the methods section of the paper.

We have included all raw data files as supplementary files (.xlsx). The cover letter has been revised accordingly.

We have included this data in the supplementary files and have indicated the location in the main text.

All supplemental files have been given captions and titles and are updated in the manuscript accordingly. 

 To our knowledge, we have not cited any retracted papers. 

Reviewers' comments:

Reviewer's Responses to Questions

Comments to the Author

1. Does the manuscript report a protocol which is of utility to the research community and adds value to the published literature?

Reviewer #1: Yes

2. Has the protocol been described in sufficient detail?

Descriptions of methods and reagents contained in the step-by-step protocol should be reported in sufficient detail for another researcher to reproduce all experiments and analyses. The protocol should describe the appropriate controls, sample sizes and replication needed to ensure that the data are robust and reproducible.

Reviewer #1: Yes

3. Does the protocol describe a validated method?

Reviewer #1: Yes

4. If the manuscript contains new data, have the authors made this data fully available?

Reviewer #1: Yes

5. Is the article presented in an intelligible fashion and written in standard English?

Reviewer #1: Yes

6. Review Comments to the Author

Reviewer #1: A good technical manuscript with a few obvious typos. The message nicely confirms well-known concepts in terms of accuracy of OD vs cell count, the influence of cell size, etc.

Thank you for your feedback. We have revised according to journal requirements and fixed typos throughout.

7. PLOS authors have the option to publish the peer review history of their article (what does this mean?). If published, this will include your full peer review and any attached files.

Do you want your identity to be public for this peer review? For information about this choice, including consent withdrawal, please see our Privacy Policy.

Reviewer #1: No

---

## [Editor Report · Decision Letter 1]

28 Sep 2022

Estimating microbial population data from optical density

PONE-D-22-14390R1

Dear Dr. Portia Mira,

We’re pleased to inform you that your manuscript has been judged scientifically suitable for publication and will be formally accepted for publication once it meets all outstanding technical requirements.

Kind regards,

Abdelwahab Omri, Pharm B, Ph.D, Laurentian University

Academic Editor

PLOS ONE

---

## [Editor Report · Acceptance letter]

5 Oct 2022

PONE-D-22-14390R1 

Estimating microbial population data from optical density 

Dear Dr. Mira:

I'm pleased to inform you that your manuscript has been deemed suitable for publication in PLOS ONE. Congratulations! Your manuscript is now with our production department. 

Kind regards, 

on behalf of

Dr. Abdelwahab Omri 

Academic Editor

PLOS ONE